# Mitochondrial Pyruvate Carrier Function in Health and Disease across the Lifespan

**DOI:** 10.3390/biom10081162

**Published:** 2020-08-08

**Authors:** Jane L. Buchanan, Eric B. Taylor

**Affiliations:** 1Department of Molecular Physiology and Biophysics, University of Iowa Carver College of Medicine, Iowa City, IA 52240, USA; Jane-Buchanan@uiowa.edu; 2Holden Comprehensive Cancer Center, University of Iowa Carver College of Medicine, Iowa City, IA 52240, USA; 3Fraternal Order of Eagles Diabetes Research Center (FOEDRC), University of Iowa Carver College of Medicine, Iowa City, IA 52240, USA; 4Abboud Cardiovascular Research Center, University of Iowa Carver College of Medicine, Iowa City, IA 52240, USA; 5Pappajohn Biomedical Institute, University of Iowa Carver College of Medicine, Iowa City, IA 52240, USA

**Keywords:** mitochondrial pyruvate carrier, MPC, lifespan, pyruvate metabolism, mitochondrial transport

## Abstract

As a nodal mediator of pyruvate metabolism, the mitochondrial pyruvate carrier (MPC) plays a pivotal role in many physiological and pathological processes across the human lifespan, from embryonic development to aging-associated neurodegeneration. Emerging research highlights the importance of the MPC in diverse conditions, such as immune cell activation, cancer cell stemness, and dopamine production in Parkinson’s disease models. Whether MPC function ameliorates or contributes to disease is highly specific to tissue and cell type. Cell- and tissue-specific differences in MPC content and activity suggest that MPC function is tightly regulated as a mechanism of metabolic, cellular, and organismal control. Accordingly, recent studies on cancer and diabetes have identified protein–protein interactions, post-translational processes, and transcriptional factors that modulate MPC function. This growing body of literature demonstrates that the MPC and other mitochondrial carriers comprise a versatile and dynamic network undergirding the metabolism of health and disease.

## 1. Introduction

The mitochondrial pyruvate carrier (MPC) was discovered in 2012 [1,2]. However, as early as 1971, studies predicted that a mitochondrial protein transported pyruvate from the cytoplasm into the mitochondria [3,4,5]. Since its discovery, the MPC has been the subject of extensive primary literature research articles and reviews detailing key aspects of its structure, function, regulation, and diverse roles in health and disease. To date, most studies examining the MPC’s role in disease using genetically tractable systems have focused on cancer metabolism and diabetes. However, smaller yet important niches for MPC research are becoming evident, highlighting the variety of situations where the MPC controls metabolism and dependent cellular functions. Some of these include embryonic and fetal health, stem cell development, neurogenerative disease, and immune cell function.

A central theme, and the focus of this review, is that the beneficial or harmful role of the MPC in a disease or physiological state is highly specific to tissue and cell type. Cell- and tissue-dependent differences in MPC content and activity suggest that MPC function is tightly regulated as a mechanism of metabolic, cellular, and organismal control. Several recent studies provide evidence that the MPC is regulated at multiple levels by transcription factors, protein–protein interactions, and post-translational modifications. Here, we examine MPC function across the lifespan, beginning with embryonic and fetal development and ending with aging-associated neurodegeneration.

## 2. The MPC in the Triumvirate of Mitochondrial Pyruvate Metabolism

Mitochondrial pyruvate metabolism is controlled by a triumvirate of enzymes—the MPC, pyruvate dehydrogenase (PDH), and pyruvate carboxylase (PC) [6,7,8]—that together modulate many physiological and pathological processes. Pyruvate is the end product of glycolysis, and its metabolism is instrumental in managing carbohydrate loads, producing ATP, and maintaining blood glucose levels. When pyruvate metabolism is dysfunctional, as first recognized in patients with inborn errors in PDH and PC, the effects are wide-ranging. Infants with either PDH or PC genetic mutations often show high blood lactate and ammonia levels, low muscle tone and lethargy, developmental delays, seizures, and even premature death.

MPC mutations are thought to be quite rare but result in the same severe symptoms observed with PDH and PC deficiency [9,10]. The MPC gates pyruvate entry into mitochondria and thus is a pivotal control point for mitochondrial pyruvate metabolism. The mammalian MPC is a complex formed by the MPC1 and MPC2 paralogs that are both necessary to stabilize the other to form a functional MPC [1,2]. Once pyruvate is inside mitochondria, the PDH complex decarboxylates it into acetyl-CoA for forward tricarboxylic acid (TCA)-cycle oxidation and ATP production, or PC carboxylates it into oxaloacetate to support biosynthesis, including nucleotide and amino acid synthesis, de novo lipogenesis, and gluconeogenesis. Not surprisingly, increases or decreases in MPC activity correspond to shifts toward oxidative or glycolytic metabolism, respectively. These metabolic shifts are crucial to support changes in cellular state, and their magnitude may differ depending on cellular state. 

## 3. The MPC in Development

### 3.1. Embryonic and Fetal Health

Due to the energetic, biosynthetic, and regulatory metabolic demands of rapidly dividing cells, MPC deficiency in utero has severe consequences for the developing fetus. Homozygous *Mpc1* or *Mpc2* deletion in mice results in embryonic lethality within the first three weeks of development [11,12,13]. In vitro *Mpc1* silencing in mouse oocytes also significantly impairs maturation [14]. Hypomorphic *Mpc1* mice exhibit early perinatal lethality, while a less severe N-terminal Mpc2 hypomorph allows for normal mouse development [11,13]. In the first study to describe inborn errors of MPC function in humans, an infant patient presented with lactic acidosis, low muscle tone, and brain abnormalities [10]. Biochemical studies of the patient’s fibroblasts revealed a defect in pyruvate transport, which contributed to the molecular identification of the MPC several years later [1,2].

In a more recent study, Oonthonpan et al. investigated the biochemical mechanisms by which patient *MPC1* C289T and T236A mutations inactivate the MPC complex [9]. *MPC1* C289T encodes a mis-spliced, truncated protein that is nonfunctional and a full-length R97W point mutant that is less stable but can form pyruvate transport-competent complexes with MPC2. Adding *MPC1* C289T to *Mpc1* knockout mouse C2C12 cells led to no detectable mis-spliced, truncated MPC1 protein and, compared to wild-type MPC1, decreased levels of MPC1 R97W. The other patient mutation investigated, *MPC1* T236A, encodes a full-length L79H point mutant that produces a stable MPC complex with protein levels similar to wild-type MPC complexes. However, MPC complexes with MPC1 L79H are not able to transport pyruvate. The patient with the MPC1 R97W mutation died at 19 months, while three patients with the MPC1 L79H mutation exhibited varying degrees of neurological and cognitive deficits. Notably, mouse studies indicate that alterations in maternal diet may be able to partially compensate for offspring MPC deficiencies in utero. For example, a ketogenic diet [12] or maternal fasting [15], which shifts metabolic substrate use from glucose and thus pyruvate to greater use of either fatty acids or ketones, attenuates some defects in pyruvate metabolism, such as lactic acidosis, in mice with *Mpc1*-mutated fetuses. A ketogenic diet also rescues many of the developmental defects observed in utero. These studies illustrate the importance of the MPC in embryonic and fetal development and suggest a potential non-pharmacological, diet intervention for MPC insufficiency and other inborn errors in pyruvate metabolism in utero.

### 3.2. Stem Cells

Proper stem cell function and growth are essential for embryonic, fetal, and postnatal development. In adults, although stem cell activity progressively decreases across the lifespan, stem cells remain important for regeneration and repair of many tissues [16,17]. Whether in embryonic or adult tissues, stem cells exist in an undifferentiated, quiescent state until an activating event (such as implantation in the uterus or stress/injury in the adult) induces a transition to proliferation. Growth is initiated and maintained by increased glycolysis, which supports nucleotide, protein, and lipid synthesis. As stem cells differentiate into specific cell types, their metabolism shifts toward oxidative phosphorylation (OXPHOS) [18]. These metabolic shifts supporting pluripotency or differentiation are controlled by MPC activity and pyruvate utilization. For example, in *Drosophila* intestinal epithelial cells, MPC loss-of-function increases stem cell proliferation, while overexpression of the MPC in epithelial cells suppresses stem cell proliferation [19,20]. Similarly, *MPC1* deletion in intestinal stem cells promotes proliferation [20], while deletion in hair follicle stem cells increases lactate-driven acceleration of the hair cycle [21]. These studies suggest that increased glycolysis due to MPC loss in epithelial or stem cells supports stem cell expansion and prevents differentiation.

Stem cells also play an important role in cancer progression and resistance to drug treatment (see Cancer section). Many cancers exhibit partial to complete loss of MPC expression that is associated with increased cell proliferation, metastasis, and stem cell marker expression [22]. MPC1 is necessary for intestinal stem cell differentiation in zebrafish, and loss of the tumor suppressor Adenomatous Polyposis Coli (APC) downregulates MPC expression to favor stem cell growth over differentiation [23]. Overall, these studies indicate that MPC activity mediates metabolic programming to promote pluripotency or differentiation during embryonic development, adult tissue repair, and tumor growth.

## 4. The MPC in Post-development Health and Disease

### 4.1. Cancer

Cancer is a disease that is characterized by abnormal cellular growth and invasion into surrounding tissues. Cancer can occur throughout the human lifespan, but prevalence increases with age and most patients receiving the diagnosis are older than 60. Similar to stem cells, many cancers support growth demands by upregulating glycolysis and channeling glycolytic intermediates into biosynthetic and reducing power-generating pathways [24]. Loss of the MPC initiates or promotes aerobic glycolysis, and in a variety of cancers, MPC disruption correlates with increased growth, metastasis, and poor survival. Nevertheless, some cancers exhibit increased MPC expression and predominantly rely on oxidative phosphorylation to support growth. This difference could be due to the metabolism of surrounding tumor tissue, tissue type, the oxygenation and blood supply of the tumor microenvironment, and the accessibility of immune cells, all of which affect the energy metabolism of a tumor and can determine whether the MPC is a mediator of pro- or anti-cancer effects.

#### 4.1.1. MPC Disruption Promotes Cancer Progression

A wealth of recent studies show that the loss of MPC activity or expression promotes cancer cell progression in diverse tissues (Table 1). In colon cancer cell lines, knockdown of *MPC1* or *MPC2* promotes loss of cell–cell polarity, increases migration capacity, and drives resistance to radiation therapy [25]. In colon cancer cell lines with low MPC protein levels, ectopic expression of the MPC impairs colony formation in soft agar, decreases xenograft growth in mice, and reduces stem cell marker expression [22]. These studies, although mostly performed in cell culture, suggest that MPC disruption in colon cancer increases growth by inducing aerobic glycolysis. Downregulation of the MPC may also promote colon cancer survival by decreasing production of reactive oxygen species (ROS), making tumor cells less prone to apoptosis by interferon-γ, an important anti-tumor cytokine [26]. Induction of the epithelial–mesenchymal transition (EMT) is associated with MPC loss and increased glutamine metabolism and may mediate increased migration capacity and metastasis. The increase in stemness associated with MPC loss, noted by Schell et al., may explain the increased resistance to radiation therapy, since stem cells are able to self-renew indefinitely [27]. Although the mechanisms by which the MPC controls cancer cell stemness are not fully understood, a noted possibility is by modulating epigenetic programming. The MPC may drive stemness through decreased cytoplasmic acetyl-CoA production dependent on and downstream from mitochondrial pyruvate uptake, resulting in less histone acetylation and preventing differentiation [20].

Another possible explanation for increased stem cell marker expression in colon cancer is that MPC loss in a stem-like cancer cell may drive expansion of the stem-like cell compartment and facilitate a shift in tumor metabolism from an OXPHOS state toward a glycolytic state (Figure 1). Although stem-like cancer cells normally make up approximately 1 to 3% of tumor cells, MPC loss may increase stem-like cell proliferation, resulting in glycolytic, stem-like cells comprising a greater fraction of the tumor. Thus, whether MPC loss occurs in a stem-like cell or differentiated cell may shape the tumor’s metabolic status and whether the tumor’s growth is enhanced or inhibited in MPC disruption studies. This paradigm illustrates how decreased MPC function in tumors may promote growth and invasion into adjacent tissue. However, a recent study suggests that MPC loss precedes and facilitates tumor formation in colon cancer [28]. In this study, using chemical and genetic colon cancer mouse models, *Mpc1* deletion in intestinal stem cells increased tumor formation and produced higher-grade tumors than *Mpc1*-positive stem cells. Conversely, in a *Drosophila* hyperproliferation model, overexpression of *Mpc1* in intestinal stem cells was sufficient to prevent tumor formation. Taken together, these studies [22,28] suggest that MPC loss in intestinal stem cells supports a glycolytic phenotype that enables tumorigenesis, and MPC loss in stem-like colon cancer cells supports increased glycolysis that drives tumor stemness and growth.

As with colon cancer, in prostate cancer cell lines, *MPC1* knockout or chemical inhibition increases invasiveness, chemotherapy resistance, and stem cell marker expression [29,30]. Increased *MPC1* or *MPC2* expression in prostate cancer patients predicts more favorable outcomes [31]. Conversely, downregulation of *MPC1* expression by chicken ovalbumin upstream promoter-transcription factor II (COUP-TFII) leads to increased invasiveness and prostate cancer cell growth in culture [32]. In prostate cancer patients, low *MPC1* and high *COUP-TFII* expression were associated with metastasis [32]. However, in this study, knockdown and overexpression of *MPC1* had no effect on the protein expression of MPC2. Given that MPC1 is needed to form fully stable MPC complexes, this is surprising and may warrant further investigation to determine if the MPC is uniquely regulated in the systems utilized in this study.

In brain cancer, patient data from The Cancer Genome Atlas (TCGA) show that low *MPC1* expression in glioblastoma correlates with worse patient survival and resistance to chemotherapy [33]. Similarly, a TCGA analysis of isocitrate dehydrogenase (IDH)-mutant gliomas associates increased *MPC1* expression with overall survival [34]. Increased *MPC1* expression via *COUP-TFII* inhibition slows growth of human glioblastoma cell lines and decreases tumor volume in xenograft studies with immune-deficient mice [35]. Although these studies suggest that MPC disruption in glioblastoma exacerbates tumor aggressiveness, studies of MPC inhibition or knockout in glioblastoma mouse models would strengthen these findings. Interestingly, in human glioblastoma patients undergoing tumor resection and in an orthotopic mouse model of human glioblastoma, ^13^C glucose tracing reveals an increase in pyruvate oxidation and glucose-derived glutamine compared to normal adjacent tissue, consistent with enhanced MPC activity [36,37]. There is also clinical evidence that a ketogenic diet may increase progression-free and overall survival for glioblastoma patients by reducing glucose availability for tumors [38]. These clinical studies [36,37,38] suggest that glioblastoma tumors exhibit increased pyruvate metabolism in vivo, which is difficult to resolve with the low levels of MPC expression observed in TCGA, cell culture, and xenograft studies [33,34,35]. Overall, more in vivo studies testing the direct contribution of the MPC to glioblastoma progression are needed to understand the role of pyruvate oxidation in brain cancer.

In kidney cancer, protein and mRNA levels for *MPC1* and *MPC2* are lower in advanced renal cell carcinoma (RCC) tumors compared to normal adjacent tissue [39], and decreased *MPC1* expression in RCC correlates with worse survival outcomes [40]. *MPC1* knockdown and chemical MPC inhibition in RCC cell lines increase migration and invasion, while RCC xenograft tumors with *MPC1* knockdown grow more rapidly than xenograft tumors from RCC cells that were transfected with a scrambled short-hairpin RNA [40]. In esophageal cancer, MPC chemical inhibition increases invasiveness and resistance to chemo/radiotherapy [41]. This same study also found that decreased *MPC1* expression in esophageal squamous cell cancer correlates with worse patient prognosis.

Lastly, in lung cancer, low *MPC1* expression in lung adenocarcinoma patient samples correlates with worse prognosis [42]. Functional experiments from the same study show that *MPC1* knockdown increases the volume of lung adenocarcinoma tumorspheres and stem cell marker expression. Similar to experiments in glioblastoma patients, ^13^C glucose tracing in non-small cell lung cancer patients (including adenocarcinoma) reveals increased glycolysis and pyruvate oxidation in tumor tissue [43]. Lactate production is also increased, consistent with the idea that circulating lactate may supply tumors with a source of pyruvate for the TCA cycle [44]. Lactate-derived, cytosolic pyruvate requires the MPC to enter the mitochondrial matrix; however, there is evidence that lactate may directly enter the mitochondrial matrix in cultured lung cancer cells, bypassing the MPC [45]. These studies highlight the complexity of elucidating the MPC’s role in lung cancer and raise the possibility of other gatekeepers of pyruvate metabolism, such as a potential inner mitochondrial membrane lactate transporter [45,46,47,48].

Overall, these studies provide human epidemiological data that associate decreased MPC expression with worse patient prognosis and animal/ cell culture data that suggest MPC loss supports aerobic glycolysis, cancer cell proliferation, and metastasis. In MPC-deficient lung, colon, and prostate cancers, an increase in stem cell markers suggests that MPC loss may increase resistance to chemo/radiation therapy and expand the stem cell compartment to shift tumor metabolism toward a growth-promoting, glycolytic state.

#### 4.1.2. MPC Disruption Inhibits Cancer Progression

Although MPC loss often promotes cancer progression, cancers that rely on mitochondrial pyruvate utilization to maintain growth or spare glutamine for glutathione production are negatively impacted by MPC disruption (Table 1). For example, although MPC inhibition drives cancer progression in most studies of prostate cancer, MPC deletion in models of androgen receptor-driven prostate cancer abolishes cell growth and is rescued by pyruvate supplementation [49]. This is likely because androgens mediate MPC expression and increase pyruvate oxidation and lipogenesis in androgen-receptor-positive prostate cancer. Androgen receptor signaling appears to be a primary driver of prostate cancer progression, with the majority of prostate cancer deaths occurring from androgen-receptor-positive, castrate-resistant prostate cancer, making the MPC an attractive therapeutic target for this prostate cancer subtype.

In liver cancer, the MPC’s role may depend on tumor genetics. Tompkins et al. show that liver-specific *Mpc1* knockout in mice impairs chemically induced hepatocellular tumorigenesis by diverting glutamine into the TCA cycle and away from glutathione synthesis [50]. This pro-tumorigenic role for the MPC is consistent with TCGA data showing hepatocellular carcinomas (HCCs) (and prostate cancers) to be the highest MPC-expressing human cancers, with MPC downregulation being a rare event [50]. Conversely, Kim et al. report that protein interactions between PUMA, a p53-controlled mitochondrial protein, and the MPC complex disrupted MPC function in human HCC cell lines and promoted tumorigenesis [51]. The apparent discrepant findings between these two studies might be explained by the p53 mutation status of the HCC models. All HCC studies in the work by Kim et al. contain wild-type p53, since p53 mediates PUMA and therefore MPC function. Conversely, because the chemical model utilized by Tompkins et al. induces random genetic mutations to recapitulate the genetic heterogeneity of human HCC, it may involve mutated p53.

In breast cancer, several studies show that chemical or estrogen-related receptor alpha (ERRα)-mediated disruption of the MPC inhibits proliferation in cell lines [52,53]. In gall bladder cancer, increased *MPC1* expression due to overexpression of peroxisome proliferator-activated receptor-gamma coactivator 1-alpha (PGC-1α) promotes metastasis in vitro and in vivo, which may increase OXPHOS and reverse the Warburg effect [54]. Lastly, in cervical cancer, chemical inhibition of the MPC decreases growth of the SiHa cervical cancer cell line [55].

Overall, these studies highlight the importance of tissue type and suggest that metabolic wiring of breast, gallbladder, and cervical cancers differs intrinsically from that of colon, brain, lung, esophageal, and kidney cancer. The requirement for certain hormone and growth factor receptors may also play a role in determining the importance of MPC expression in tumors. Thus far, the idea that MPC disruption downregulates cancer metabolism and impairs tumorigenesis has two limitations: first, only a small number of studies have investigated the role of the MPC in breast, gallbladder, and cervical cancer; second, several studies focused on overexpression or inhibition of transcription factors that likely have effects other than altered MPC expression. Overall, cancers that appear to rely on OXPHOS to maintain growth or spare glutamine for glutathione production are adversely affected by MPC disruption. Other tissue-specific factors that support the MPC’s role as a pro-cancer mediator are the presence of androgen receptors in prostate cancer and the potential for mutant p53 in liver cancer.

### 4.2. Pathology Associated with Immune Cell Function

The role of the MPC in immune cell dysfunction is a new area of research for the MPC field. Similar to many cancer cells, T cells increase aerobic glycolysis during activation and expansion to support the metabolic demands of rapid proliferation and cytokine production [79,80,81]. Aging-associated inflammation and immune dysfunction are thought to be a consequence of inappropriate T cell expansion. In support of this idea, *Mpc1* deletion in T cells increases the pool of activated T cells in aging mice [82]. *Mpc1* deletion in hematopoiesis also leads to a larger proportion of activated T cells, increasing the probability of autoimmune encephalitis in mice [83]. However, *Mpc2* deletion in long-lived plasma cells increases cell death and loss of vaccine-specific antibodies [84]. The importance of pyruvate-dependent respiration in long-lived plasma cells may be because long-lived plasma cells survive for years after infection or vaccination in the bone marrow, whereas activated T cells and short-lived plasma cells undergo apoptosis after a couple of days [85,86,87,88]. Therefore, MPC expression in immune cells may depend on their need to rapidly expand in response to infection or to live in a prolonged, quiescent state across the human lifespan.

### 4.3. Pathology Associated with Retinal and Visual Function

The retina is an essential component of the human eye, consisting of multiple layers of neuronal and light-sensitive tissue that relay visual images to the brain through electrical signaling. These tissue layers exhibit metabolic synergy in numerous ways. For example, the rods and cones of the photoreceptor layer perform aerobic glycolysis and provide the adjacent retinal pigment epithelium (RPE) with lactate [89,90], and the RPE synthesizes amino acids, such as glutamine and glutamate, to support photoreceptor metabolism and function [91]. Interestingly, although retinal pyruvate oxidation is minimal [89,90], the MPC appears to be essential for photoreceptor integrity and visual function [92]. Retinal-specific *Mpc1* knockout leads to photoreceptor degeneration, potentially by limiting TCA-cycle-derived acetyl-CoA and non-essential amino acids needed for daily biosynthesis of rod outer segments [92]. This study also suggests that high levels of glutamine-derived aspartate may contribute to the progressive decline of visual function in *Mpc1* knockout mice by depleting photoreceptors of glutamate, which is required for synaptic transmission. This finding corroborates a previous study showing that zaprinast, a lead but failed compound in sildenafil (Viagra) development, is a potent MPC inhibitor that induces retinal aspartate accumulation at the expense of glutamate [93]. Overall, the MPC appears to be an integral mediator of retinal structure and visual function. However, given the differential metabolism of retinal layers and the potential confounding effects of constitutive MPC knockout on retinal development, more work remains to understand the MPC’s global role in the retina.

### 4.4. Diabetes

Diabetes mellitus is a metabolic disease that is categorized into type 1 diabetes (T1D) and type 2 diabetes (T2D). T1D onset usually occurs in childhood as a result of autoantibodies that destroy pancreatic beta cells and thus the ability to produce insulin. T2D affects children and adults, although it most commonly manifests during middle age. T2D is characterized by obesity, insulin resistance, and hyperinsulinemia, which result in decreased tissue glucose uptake and increased hepatic gluconeogenesis. Together, these factors drive chronically elevated blood glucose (hyperglycemia). Hyperglycemia is particularly detrimental to nervous tissue and vasculature. T2D patients can exhibit peripheral nerve damage, chronic kidney disease, glaucoma, and vision problems and have increased risk of stroke and hypertension. Whether the MPC ameliorates or contributes to T2D pathology predominantly depends on the tissue in question. Liver and skeletal muscle MPC disruptions improve aspects of T2D pathology, such as hyperglycemia, hepatic inflammation, and obesity. Conversely, in pancreatic beta cells, MPC loss impairs glucose-stimulated insulin secretion and worsens T2D. MPC disruption in the heart and kidney may exacerbate diabetes-mediated damage, although studies are limited in number and the mechanisms are less clear.

#### 4.4.1. MPC Disruption Improves Glycemia and Attenuates Diabetes-Related Pathology

MPC loss or inhibition in skeletal muscle and liver improves T2D pathology. Human myocytes acutely increase glucose uptake after treatment with thiazolidinediones (TZD), a class of drugs that augments peroxisome proliferator-activated receptor gamma (PPAR-γ) activity and acutely inhibits the MPC in biochemical and cell-based assays [56]. Similarly, *Mpc1* knockout in skeletal muscle increases glucose uptake, fat oxidation, and whole-body insulin sensitivity while preserving lean mass during recovery from obesity (Figure 2) [57]. In these studies, MPC disruption causes pyruvate to accumulate in the cytosol, where it is converted to lactate and exported from myocytes [56,57]. It is possible that increased alanine is also excreted; however, this has not been experimentally validated. In vivo, lactate is taken up by the liver for gluconeogenesis. Because gluconeogenesis requires more ATP than glycolysis produces, fat oxidation is increased to provide reducing equivalents for net ATP synthesis [57]. Loss of body fat improves insulin sensitivity and promotes glucose uptake by muscle. Consequently, this may explain the whole-body leanness and lower blood glucose levels observed in skeletal muscle *Mpc1*-knockout mouse studies.

In the liver, *Mpc1* or *Mpc2* knockout decreases hepatic gluconeogenesis and attenuates hyperglycemia in high-fat-diet-induced or leptin-receptor-deficient (*db/db)* mice without causing hypoglycemia in lean, normal chow-fed mice (Figure 3) [58,59,60,61]. These studies demonstrate that liver MPC disruption inhibits pyruvate transport into hepatocyte mitochondria and disrupts gluconeogenesis, likely by impairing pyruvate metabolism to oxaloacetate by pyruvate carboxylase, a key initial step in gluconeogenesis. The drug berberine also decreases hepatic gluconeogenesis in high-fat-diet-induced mice and correlates with decreased MPC protein via sirtuin 3 (SIRT3) deacetylation [62]. Genetic liver MPC disruption in mice also attenuates liver fibrosis and inflammation, which are hallmarks of nonalcoholic fatty liver disease (NAFLD) and nonalcoholic steatohepatitis (NASH) [59,64]. *Mpc1* knockout in hepatocytes improves NAFLD [50], while MPC inhibition with pioglitazone (a TZD), MSCD-0602 (a PPAR-γ-sparing, TZD-like molecule) or liver *Mpc2* deletion correlates with improved NASH [63,64]. MPC disruption may improve NASH/NAFLD phenotypes by decreasing the amount of pyruvate that is metabolized to acetyl-CoA, an essential substrate for de novo lipogenesis, and/or by decreasing mitochondrial ROS production, which is associated with reduced hepatic inflammation [59]. Notably, NAFLD and NASH have been shown to involve excessive TCA cycle flux, which degrades control of mitochondrial oxidative capacity [94]. Blocking pyruvate entry into the mitochondria likely helps to ease this flow of carbon, which has been shown with liver *Mpc1* deletion in NAFLD [59] and with TZD treatment in NASH [95].

Pharmacologic inhibition of the MPC may be useful for treating T2D, NAFLD, and NASH. MSDC-0602, a PPAR-γ-sparing, TZD-like molecule that directly inhibits the MPC, may reduce hepatic gluconeogenesis in vivo similar to genetic MPC disruption by obstructing access of pyruvate to pyruvate carboxylase for metabolism to the gluconeogenic precursor, oxaloacetate [60]. In a phase IIb clinical trial, T2D patients receiving another PPAR-γ-sparing, TZD-like molecule with MPC-inhibiting activity, MSDC-0160, showed decreased glycated hemoglobin (HbA1c) compared to placebo after 12 weeks [96]. A more recent phase IIb trial showed that MSDC-0602K decreases blood glucose, glycated hemoglobin, insulin, liver enzymes, and NAFLD activity scores compared to placebo [97]. However, since trace but biologically relevant residual PPAR-γ agonism of these TZD-like molecules is challenging to account for in vivo, improvements in diabetes parameters and NASH cannot yet be fully attributed to MPC inhibition.

Although most of the literature provides evidence that T2D symptoms improve with MPC loss in skeletal muscle and liver, there are a few studies that report opposing changes associated with MPC alterations or studies that have utilized mice heterozygous for either full-body *Mpc1* or *Mpc2* deletion. While the nature of these studies makes results more difficult to interpret, they are mentioned here for the sake of completeness. In striated muscle, inactivation of E4 transcription factor 1 (E4F1), a regulator of several pyruvate oxidation genes, decreases *MPC1* expression and pyruvate dehydrogenase (PDH) activity and correlates with lactic acidemia and endurance defects [98]. However, these effects abated with chemical stimulation of pyruvate dehydrogenase, suggesting that the phenotype was driven predominantly by changes in PDH activity and not the MPC. In another study, after 14 weeks of calorie-restricting middle-aged mice, skeletal muscle MPC content increased and age-related muscle loss decreased [99]. However, calorie restriction alters metabolic programming other than the MPC, muddying interpretations for the role of the MPC in this study. In heterozygous *Mpc1* knockout models, female mice were reported to exhibit decreased fertility and increased body weight [66], which contrasts with a study by Zou et al. where male mice had decreased body weight [65]. Whether these studies reflect generalizable sex-specific effects of partial MPC disruption or idiosyncrasies of the individual mouse lines used remains to be determined.

#### 4.4.2. MPC Disruption Exacerbates Diabetes-Related Pathology

Decreased pyruvate utilization due to MPC loss may worsen T2D pathology in pancreatic beta cells, kidney, and heart. Islet or beta cell *Mpc2* knockout mice and *Mpc1* knockout in *Drosophila* resulted in impaired glucose-stimulated insulin secretion and elevated glycemia when challenged with a glucose bolus [70]. Similarly, N-terminally truncated, hypomorphic Mpc2 protein expression due to a global *Mpc2* mutation decreased glucose-stimulated insulin release in mice [11]. *MPC1* and *MPC2* silencing via siRNAs in beta cell lines and rat and human islets also decreased glucose-stimulated insulin secretion [71]. These studies indicate that MPC disruption in pancreatic beta cells impairs glucose-stimulated insulin release. Interestingly, in the absence of a high glucose challenge, these animal models did not display overt diabetes or develop insulin resistance [70]. Overall, these studies fit a straightforward model of how pancreatic beta cells sense glucose, in part through mitochondrial pyruvate oxidation, to determine appropriate levels of insulin secretion (Figure 4).

In a high-glucose cell culture model designed to mimic T2D hyperglycemia, MPC chemical inhibition and siRNA knockdown of *MPC2* in podocytes induced mitochondrial damage and increased apoptosis [67]. Alternatively, the drug artemether protected against diabetic kidney disease in a T2D (*db/db*) mouse model and was associated with increased MPC content [68]. Renal tubule MPC1 and MPC2 protein expression were significantly lower in diabetic nephropathy patients compared to patients with non-diabetic kidney disease [69]. These studies raise the possibility that decreased MPC expression contributes to worsened pathology in the diabetic kidney. However, the first study was performed in a cell culture model, potentially limiting its applicability to in vivo processes, the second study uses a drug that has physiological effects other than increased MPC content, and the third study is also correlational. In the heart, decreased MPC activity via acetylation of Mpc2 lysine 19 and 26 in Akita T1D mice correlates with diabetic cardiomyopathy [72]. However, this is only one associative study in a T1D model. Further in vivo studies that assess the effects of kidney and heart MPC knockout on diabetes pathology are needed before more definitive conclusions can be made.

### 4.5. Neurogenerative Diseases

Neurodegenerative diseases such as Parkinson’s disease (PD) and Alzheimer’s disease (AD) involve progressive neuronal dysfunction and death and predominantly affect adults near the end of the human lifespan. PD is characterized by the degeneration of dopamine-producing neurons in the substantia nigra region of the brain, resulting in gradual loss of motor function and eventually aspects of cognitive function. AD is characterized by intracellular tau protein aggregation and extracellular beta-amyloid protein aggregation in the cortical regions of the brain, resulting in the decline of executive cognitive functions, social functions, and ultimately death. Early findings suggest the role of the MPC in these diseases can be protective or pathogenic depending on the disease and the model utilized.

#### 4.5.1. MPC Disruption is Protective in Neurodegenerative Disease

Several studies indicate that reduced mitochondrial pyruvate utilization is protective in Parkinson’s and Alzheimer’s disease. Chemical inhibition of the MPC in a chemically induced PD mouse model increased survival of substantia nigra dopaminergic neurons and augmented striatal dopamine production [76]. This same study also used a genetic PD mouse model and concluded that MPC inhibition with MSDC-0160 decreased neuroinflammation and improved motor function. Although MSDC-0160 is a TZD-like molecule that is thought to have PPAR-γ-sparing effects, it is possible that weak PPAR-γ activity could have contributed to the decreases in neuroinflammation. Chemical inhibition of the MPC in cultured neurons reduced neuronal death from glutamate-induced excitotoxicity, a phenotype that is associated with AD [73]. By decreasing mitochondrial pyruvate utilization, mitochondrial glutamate oxidation is adaptively increased to sustain TCA cycle activity, decreasing the amount of glutamate available for synaptic release.

#### 4.5.2. MPC Disruption Contributes to or does not Affect Neurodegenerative or Psychiatric Disease

There are also studies showing that MPC disruption may worsen pathology in schizophrenia and Alzheimer’s disease. An intronic *MPC2* mutation was positively and significantly correlated with schizophrenia in East Asian populations [77]; however, an earlier genome-wide association study (GWAS) study of the same single nucleotide polymorphism (SNP) did not find a significant association [78], possibly due to differences in biostatistical methodology. In an in vitro excitotoxicity model that may inform AD, chemical inhibition of the MPC eliminated the neuroprotective effect of lactate supplementation on glutamate-induced excitotoxicity [74]. These findings differ with the study from Divakaruni et al., in which chemical inhibition of the MPC preserved neuronal viability during glutamate-induced excitotoxicity. Both studies used the same MPC inhibitor, UK5099, and the same concentration of glutamate in culture. However, a key difference is that Jourdain et al. assessed cell viability 2.5 h after treatment with 1 uM UK5099, whereas Divakaruni et al. assessed viability 24 h after treatment with 10 uM UK5099. The differences in cell culture models may also explain the opposing study findings. Divakaruni et al.’s model included a complete media with non-glucose substrates, which would enable a metabolic switch to alternative substrates to support OXPHOS after blocking the MPC. Conversely, Jourdain et al. used artificial cerebrospinal fluid supplemented with glucose as the sole carbon fuel, which did not provide cells with alternative mitochondrial fuels when the MPC was inhibited. Lastly, another study showed that decreased MPC2 protein levels and destabilization of MPC complexes in familial AD cell models are associated with decreased ATP levels, a hallmark of AD pathology [75]. Overall, these studies suggest that MPC disruption can be neuroprotective or neuropathic depending on context. They also raise larger questions about how media composition in ex vivo assays impacts MPC disruption.

## 5. Regulation of MPC Expression or Activity

Recent studies in cancer and diabetes have elucidated transcription factors, protein interactions, and post-transcriptional modifications that can control MPC expression and function. Several transcription factors have been shown to regulate *MPC1/2* expression in various tissues and diseases. In cancer, upregulation of chicken ovalbumin upstream promoter-transcription factor II (COUP-TFII) transcript expression downregulates *MPC1* expression to drive cell growth and metastasis in prostate cancer and glioblastoma [32,35]. In androgen-receptor-positive prostate cancers, androgen receptor binding to the intron of the *MPC2* locus directly controls transcription of *MPC2* [49]. In renal cell carcinoma, PGC-1α regulates *MPC1* transcription by recruiting ERRα to the ERRα response element 2 on the *MPC1* promoter [39]. Modulation of MPC expression via PGC-1α or ERRα also occurs in cholangioma and certain breast cancers [53,54]. To facilitate hepatic gluconeogenesis during fasting, glucagon increases *MPC1* transcription through recruitment of cAMP-responsive element-binding protein (CREB) to the *MPC1* promoter in hepatocytes [58].

Certain protein interactions may also modulate MPC activity. In wild-type p53 hepatocellular carcinoma, p53 disrupts MPC activity by activating transcription of PUMA, a mitochondrial protein that binds to the MPC1 protein and inhibits the oligomerization of MPC1 and MPC2 [51]. This study also found that IkB kinase beta (IKKβ), which promotes a metabolic shift to aerobic glycolysis in many cancers [100], phosphorylates PUMA at serine residues 96 and 106 to enable PUMA binding to the MPC complex and recruits PUMA from the cytoplasm to the mitochondria.

Lastly, post-translational modifications have been identified for the MPC proteins, and several have been suggested to affect MPC activity. In a T1D mouse model, hyperacetylation of Mpc2 at lysine residues 19 and 26 was associated with decreased pyruvate transport in isolated heart mitochondria despite normal expression levels of Mpc1 and Mpc2 [72]. In the same study, a K19Q/K26Q Mpc2 mutant that was designed to structurally and functionally mimic lysine acetylation was expressed in H9C2 cells, leading to decreased pyruvate oxidation. This finding suggests that acetylation at MPC2 lysine residues 19 and 26 may reduce MPC activity in the diabetic heart. Conversely, in the presence of high glucose, sirtuin 3 (SIRT3) binds to and deacetylates MPC1 at lysine residues 45 and 46 and enhances MPC1 activity [101]. Other post-translational modifications have been proposed, but their effect on MPC activity and contribution to physiology or pathology remain unclear. Overall, the current literature suggests that MPC expression can be regulated by various transcription factors and coactivators and that protein interactions or post-translational modifications can modulate MPC activity.

## 6. Conclusions and Future Directions

Although endocrinology and cancer metabolism have been focal points of early MPC research, future investigations of the MPC will likely further focus on fetal and placental development, immunology, stem cell biology, and neurology as the importance of metabolic function in these processes are realized. A key issue that has yet to be addressed is how MPC regulation is coordinated during transitions from healthy to diseased states across the human lifespan. Future studies will also likely elucidate novel connections between MPC regulation and other mitochondrial transporters, revealing a flexible, dynamic network that modulates human metabolism in health and disease [102,103].

## Figures and Tables

**Figure 1 biomolecules-10-01162-f001:**
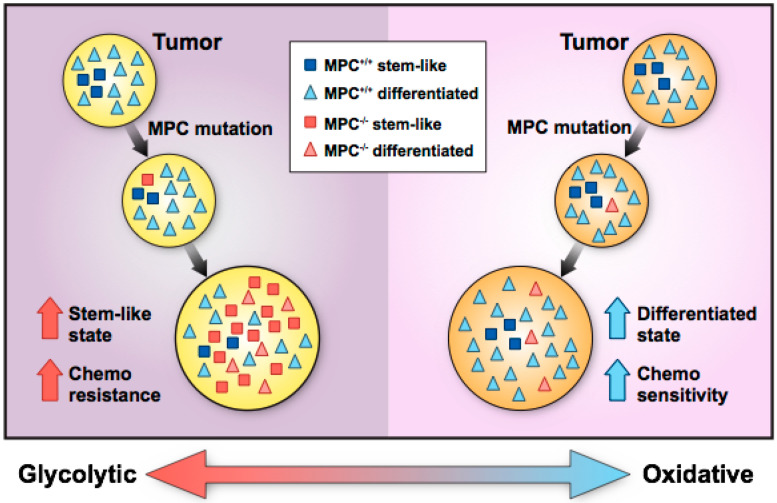
Hypothetical paradigm for the development of mitochondrial pyruvate carrier (MPC)-positive and -negative cancers. Left panel: Lack of MPC expression in stem-like cancer cells increases glycolysis and promotes proliferation, which contributes to tumor initiation and progression. Increased proliferation of MPC-deficient cells leads to a tumor enriched in glycolytic, stem-like cells that convey chemoresistance and invasiveness. Right panel: Sporadic loss of MPC expression in more differentiated, oxidative tumor cells promotes glycolysis but does not accelerate, or even impairs, cell proliferation. The tumor remains mostly oxidative with retained MPC expression, decreased invasiveness, and chemosensitivity.

**Figure 2 biomolecules-10-01162-f002:**
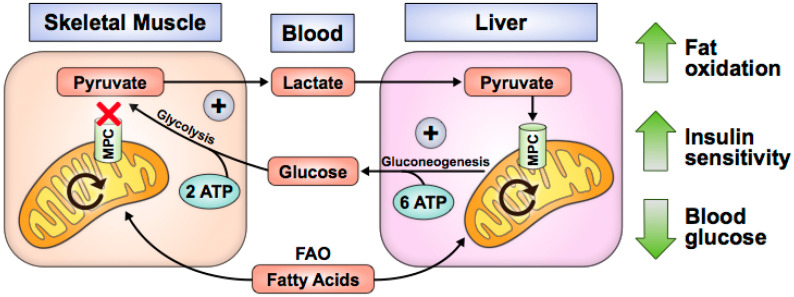
Skeletal muscle MPC disruption increases glucose uptake, fat oxidation, and insulin sensitivity. Fatty acid oxidation (FAO) provides ATP for energetically futile Cori Cycling.

**Figure 3 biomolecules-10-01162-f003:**
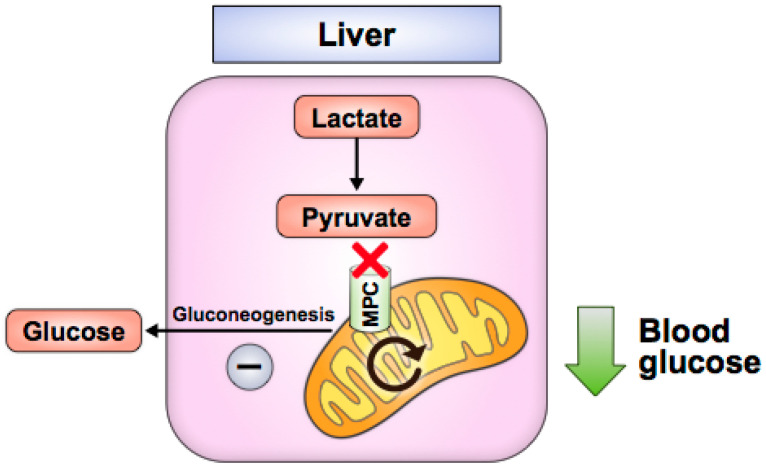
Liver MPC disruption attenuates hyperglycemia by impairing gluconeogenic pyruvate flux.

**Figure 4 biomolecules-10-01162-f004:**
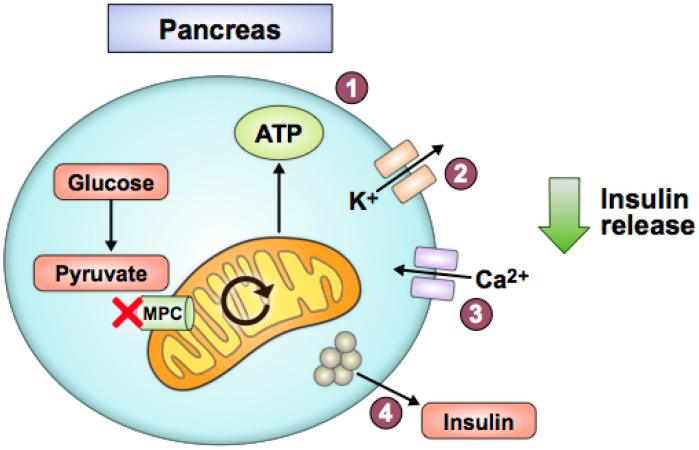
Pancreatic beta cell MPC disruption impairs glucose-stimulated insulin secretion. With decreased pyruvate-fueled mitochondrial ATP production (1), potassium-ATP channels stay open longer (2), and the cell does not depolarize as easily. Consequently, calcium influx is reduced (3), and insulin release does not occur as readily (4).

**Table 1 biomolecules-10-01162-t001:** The MPC in various disease models.

Disease and Tissue Affected	MPC in Disease Model (MPC Knockout/Knockdown = KO/KD, MPC Re-Expression/Overexpression = O, Xenograft = X, Drug Inhibition of MPC = D, MPC Expression Correlates with Patient Survival = S, MPC Expression/Protein Correlates with Disease = C, MPC Regulation = $-$$$$$)				MPC Disruption Ameliorates (+) or Exacerbates (−) Disease	Reference
	Cell Culture	Animal	Patient Database	Patient Samples		
**Cancer**						
Colon	KD				−	[25]
	O, D	X	S		−	[22]
	O				−	[26]
Prostate						
	KO				−	[29]
	D				−	[30]
				C, S	−	[31]
	KD, O		S		−	[32]
	KO, D	X	S	C	+	[49]
Ovarian	D				−	[29]
Brain			S		−	[33]
			S		−	[34]
	$				−	[35]
Kidney	D, KO			C	−	[39]
	KD, O, D	X	S	C, S	−	[40]
Esophageal	D			C, S	−	[41]
Lung	O, KD, D	X	S	C	−	[42]
Liver						
p53 null (?)	KO, D	KO			+	[50]
p53 wild-type	$$, D				−	[51]
Breast	D				+	[52]
	$$$, O				+	[53]
Gallbladder	$$$$$				+	[54]
Cervical	D	X, D			+	[55]
Pharynx	KD, D				+	[55]
**Diabetes-related Diseases**						
Skeletal muscle	KD, D				+	[56]
	D *	KO			+	[57]
Liver	KD, O, D	D			+	[58]
	D	KO			+	[59]
	KO, D	KO			+	[60]
	KO, D	KO			+	[61]
	O, D	D			+	[62]
	KO, D	KO			+	[50]
	KO, D	KO, D			+	[63]
	KO, D	KO, D			+	[64]
Whole-body		KO (het)			+	[65]
		KO (het)			−	[66]
Kidney	KD, D				−	[67]
		D		C	−	[68]
				C	−	[69]
Pancreas	KO	KO			−	[70]
		KO			−	[11]
	KD, D	D			−	[71]
Heart		$$$$			−	[72]
**Neurodegenerative Diseases**						
Alzheimer’s	D				+	[73]
	D				−	[74]
	D, O				−	[75]
Parkinson’s	D	D			+	[76]
Schizophrenia			S		−	[77]
			S		+	[78]

* Ex vivo permeabilized muscle; ^$^ MPC expression correlates with chicken ovalbumin upstream promoter-transcription factor II (COUP-TFII); ^$$^ MPC proteins interact with PUMA; ^$$$^ MPC expression correlates with estrogen-related receptor alpha (ERRα); ^$$$$^ MPC activity correlates with hyperacetylation; ^$$$$$^ MPC expression correlates with peroxisome proliferator-activated receptor-gamma coactivator 1-alpha (PGC-1α).

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
