# Peer review of "Mitochondrial Pyruvate Carrier Function in Health and Disease across the Lifespan"

_biomolecules, 2020, doi:10.3390/biom10081162_

Round 1

Reviewer 1 Report

In the review "Mitochondrial Pyruvate Carrier Function in Health and Disease Across the Lifespan" (biomolecules-887825) Buchanan & Taylor dissect the current knowledge of the involvement of the mitochondrial pyruvate carrier (MPC) in human diseases. MPC has a central role at the crossroads between mitochondrial and cytoplasmic energy metabolism due to its function in importing pyruvate, derived from glycolysis in the cytoplasm, into mitochondria to be oxidized in the tricarboxylic acid cycle. Recent studies have associated MPC function with diabetes and various types of cancers. The review is well written and it is clear that the authors have made a lot of effort to make the subject, based on many apparently contradictory studies, comprehensive. The manuscript is recommended for publication in Biomolecules after the below comments have been considered.

  1. The two mutations in MPC1 discussed in section 3.1 should be discussed in more detail because it is an interesting story and there is a clear direct link between MPC function and disease.

  1. Due to the complexity of cancers, which depend on several factors, the nature of the association with MPC function is less straight-forward in section 4.1. Where possible, could the authors speculate more clearly whether the modulated MPC function is an initial factor contributing to cause tumour formation or is it a secondary effect in cells already destined for tumourogenisis. (The referee is well aware that his request is almost impossible but the hypotheses are valuable.)

  2. Fig. 2. Maybe also "Alanine" should be indicated with "Lactate" in the figure to indicate the increased 

Author Response

1. We thank the reviewer for this suggestion, which we believe improves the manuscript by increasing human disease relevance. The section on MPC1 patient mutations in 3.1 is now expanded with more thorough discussion. We now explain in greater detail how Oonthonpan et al. investigated the effects of two patient MPC1 mutations on MPC protein content and transport function. Three MPC1 protein variants are discussed, including their protein abundance, effect on MPC transport function, and the degree to which they form stable MPC complexes. The patient mutations are then related to the disease symptoms of each patient.

2. We agree that this point is interesting and worthwhile to address.  We now discuss the MPC’s role in tumor initiation in the cancer stem cell section in 4.1.1. A 2020 study by Bensard et al. suggests that MPC deletion in intestinal stem cells facilitates a metabolic shift toward glycolysis that leads to increased tumor formation and higher-grade tumors in colon cancer mouse models. The findings of this study and others mentioned in 4.1.1 indicate that MPC loss may not only precede and facilitate tumor formation in certain cancers, but also increase tumor growth, stemness, and metastasis after the tumor has formed.

3. We agree that including alanine alongside lactate in Figure 2 would increase the conceptual completeness of the figure. However, while muscle alanine excretion may be increased with muscle MPC disruption, it has not been experimentally addressed to nearly the same extent as increased lactate excretion. Therefore, we revised the manuscript to address the possibility of increase muscle alanine excretion in lines 344-345.

Reviewer 2 Report

This is a very nice and timely review on the mitochondrial pyruvate carrier (MPC). The authors describe how this recently described carrier is involved in the physiology and pathophysiology of several key organs. The review is well illustrated and the table summarizing the involvement of the MPC in various disease models very informative. All important papers on the MPC are cited and the review will be of great interest for those working in the field of metabolism.  

Author Response

We thank the reviewer for their attention and positive comments.